# Selected Exogenous (Occupational and Environmental) Risk Factors for Cardiovascular Diseases in Military and Aviation

**DOI:** 10.3390/jcm12237492

**Published:** 2023-12-04

**Authors:** Ewelina Maculewicz, Agata Pabin, Łukasz Dziuda, Małgorzata Białek, Agnieszka Białek

**Affiliations:** 1Faculty of Physical Education, Jozef Pilsudski University of Physical Education in Warsaw, Marymoncka 34, 00-968 Warsaw, Poland; 2Military Institute of Aviation Medicine, Krasinskiego 54/56, 01-755 Warsaw, Poland; apabin@wiml.waw.pl (A.P.); ldziuda@wiml.waw.pl (Ł.D.); 3The Kielanowski Institute of Animal Physiology and Nutrition, Polish Academy of Sciences, Instytucka 3, 05-110 Jabłonna, Poland; m.bialek@ifzz.pl; 4School of Health and Medical Sciences, University of Economics and Human Sciences in Warsaw, Okopowa 59, 01-043 Warsaw, Poland

**Keywords:** aviation, cardiovascular diseases, military, pilots, soldiers

## Abstract

Cardiovascular diseases are a group of disorders of heart and blood vessels which are the leading cause of death globally. They are also a very important cause of military unsuitability especially among military pilots. Some of the risk factors cannot be modified, but a large group of them can be modified by introducing healthy habits or behaviors, such as regular physical activity, quitting of tobacco smoking, cessation of harmful alcohol consumption, or stress avoidance. Specific occupational and environmental factors, such as acceleration, pressure changes, hypoxia, thermal stress, noise, vibration, prolonged sedentary posture, difficulties in moving the limbs, shifts, work shift irregularities, circadian dysrhythmia, variations in the microclimate, motion sickness, radiation, toxicological threats, occupational stress, emotional tension, highly demanding tasks especially during take-off and landing, predominance of intellectual over physical activity, and forced operation speed against time shortage are considered as characteristic for military personnel, especially military aircrews. They are of special concern as some of them are considered as potential CVD risk factors. The aim of this study was to discuss the influence of selected occupational and environmental factors (noise, altitude, hypoxia, acceleration forces, tobacco smoking, oral health, and stress) regarding their importance for CVD risk in military professionals and military aviators. The performed revision confirmed that they are exposed to certain characteristic conditions, which may influence the CVD risk but the number of solid scientific data regarding this subject are limited. There is a great need to perform complex studies on environmental and occupational risk factors for CVDs in military personnel and military aviators as well as how to minimize their influence to prolong the state of health and military suitability of this professional groups.

## 1. Introduction

Cardiovascular diseases (CVDs) are a group of disorders of the heart and blood vessels. They include coronary heart disease, cerebrovascular, peripheral arterial disease, rheumatic heart disease, congenital heart disease, and deep vein thrombosis, as well as pulmonary embolism. According to the World Health Organization (WHO), in 2019 about 17.9 million people died from CVDs which makes CVDs the leading cause of death globally [1]. CVDs are also very important in aviation as such diseases as cardiac arrhythmias, acute coronary syndrome, pulmonary embolism, sudden cardiac death, and stroke are responsible for many in-flight incapacitations of commercial pilots [2]. Similarly, in military pilots, coronary artery disease has been recognized as a significant cause of potential in-flight incapacitation and the most common reason for permanent flying incapacitation. Sudden cardiac death has been found to be the initial presentation of coronary artery disease in 20% to even 80% of military pilots [3]. Cardiovascular and cerebrovascular incidents have been reported among the most prevalent causes of flight incapacitation in the United Kingdom and CVDs have been reported as the most frequent cause of permanent groundings among Korean airline pilots [4].

Numerous risk factors have been so far established regarding their impact on CVD risk. They can be divided into non-modifiable and modifiable risk factors. Non-modifiable CVD risk factors include age, gender, ethnicity, and family history (related to polymorphisms of certain genes). Modifiable risk factors include some endogenous modifiable risk factors, such as hypertension, hyperlipidemia, diabetes, or obesity accompanied by some novel risk factors, including inflammation, non-alcoholic fatty liver disease (NAFLD), chronic kidney disease (CKD), systemic lupus erythematosus (SLE), rheumatoid arthritis (RA), inflammatory bowel disease (IBD), human immunodeficiency virus (HIV), thyroid disease, sex hormone levels, or vitamin D levels have also been suggested as influencing CVD prevalence and development [5,6,7,8]. Their implication for CVD risk in military professionals, including military pilots, has been discussed previously, as well as the importance of certain exogenous risk factors, such as physical inactivity and diet [4,9,10,11,12,13,14,15,16,17,18,19]. Exogenous, behavioral CVD risk factors include tobacco smoking, salt consumption, fruit and vegetable dietary intake, physical inactivity, abusive alcohol consumption, or stress. Military personnel, especially military aircrew, are of special concern due to the explicit factors associated with specific occupational and environmental factors, such as acceleration, pressure changes, hypoxia, thermal stress, noise, vibration, prolonged sedentary posture, difficulties in moving the limbs, shifts, work shift irregularities, circadian dysrhythmia, variations in the microclimate, motion sickness, radiation, toxicological threats, occupational stress, emotional tension, highly demanding tasks especially during take-off and landing, predominance of intellectual over physical activity, forced operation speed against time shortage, and others (Figure 1) [20,21,22,23,24].

Some of the factors (e.g., tobacco smoking, hypoxia, altitude, stress, and different stressors) share the same pathways of the contribution to CVD development and progression, including oxidative stress and inflammation, which have special meaning for atherosclerosis, which is considered as an intermediate in CVD development. It begins at sites of endothelial injury and consists of three stages: initiation (encompassing endothelial injury and dysfunction, atherogenic lipid deposition and a proinflammatory state), progression (development of atherosclerotic plaques), and complications (limitation of myocardial blood flow or plaque rupture, which triggers clot formation and leads to the acute obstruction of the arterial lumen resulting in clinical events). Oxidative stress and inflammation are interrelated and they form a vicious cycle during atherosclerosis and CVD development and progression. Oxidative stress leads to the oxidation of LDL. The oxidative forms of LDL-C (oxLDL) are more easily taken up by macrophages and are involved in the formation of foam cells. The lipid-laden macrophages are deposited underneath the endothelium of arteries taking part in atherosclerotic plaque formation. Reactive oxygen species (ROS) also initiate expression of adhesion molecules, stimulation of vascular smooth muscle proliferation and migration, apoptosis in the endothelium, oxidation of lipids, activation of matrix metalloproteinases, and alteration of vasomotor activity. It is of special importance in case of military pilots and crew members as flying conditions (like high altitude and hypoxia) decrease oxygen availability and increase the formation of ROS. Oxidative stress activates transcription factors that alter inflammatory cytokines, soluble mediators, and chemokines. Hence, cytokines released by inflammatory cells gather inflammatory cells to the sites of inflammation, leading to increased ROS production, which confirms strong dependence between oxidative stress and inflammation. Vascular inflammation contributes to atherosclerosis by numerous mechanisms, including an increased expression of adhesion molecules and their ligands, extravasation of leukocytes, activation of pro-inflammatory signaling pathways, and an increase in oxidative stress as well as cytokine production in the arteries. Detailed interplay of oxidative stress and inflammation has been previously described [9,25,26].

The importance of mitochondria as a crossroad where oxidative stress and inflammation are interconnected in CVD progression should be mentioned. Mitochondrial oxidative stress is involved in endothelial dysfunction and apoptosis of cardiomyocytes. Also, dysregulation of the antioxidant defense system, especially disruption of antioxidant enzyme activity, contributes to mitochondrial dysfunction and is strictly involved in CVD progression. ROS production is also related to mitochondria dysfunctions as they destroy the electron transport chain, leading to further ROS formation, aggravation of mitochondrial dysfunction and cytochrome c release, and aggravation of the intrinsic apoptotic pathway in cardiomyocytes. Hypoxic conditions contribute to the development of CVDs also by the fact that mitochondria sense the diminished level of oxygen and are considered the main site of hypoxic damage. However, the influence of hypoxia acting via hypoxia-inducible factor-1 (HIF-1) is dual. Acute hypoxia leads to accumulation of HIF-1, its transfer to nuclei and activation of genes responsible for a shift of aerobic ATP production via the electron transport chain to glycolysis which is Ca^2+^-upregulated with consequent acidification of the environment, sodium overload, and breakdown of ATP production. In this condition, HIF-1 exerts cardioprotective effects by alleviating ROS generation and stimulating the removal of unwanted mitochondria through mitophagy. In chronic hypoxic conditions, HIF-1 increases ROS levels, stimulates inflammasome gene expression (including the interleukin 1ß (IL-1ß) gene), and contributes to atherosclerosis development, which in turn confirms the importance of hypoxia in CVD development [27,28,29].

Some of the environmental and occupational risk factors typical for military professional and military aviators are considered as potential CVD risk factors and they will be discussed in this paper.

## 2. Materials and Methods

A literature review was performed with the aim of answering the following research question: “What are the potential exogenous CVD’s risk factors characteristic for military and aviation?”. The following electronic databases were searched: PubMed, Scopus, and Google Scholar. Articles relevant for this study were searched using the following keywords: cardiovascular diseases, CVD, CVDs, soldiers, pilots, army, military, aviation, crew members, cardiovascular diseases risk factors, risk factors of CVD, smoking army, stress, noise, altitude, acceleration forces, G-forces, hypoxia, oral health, caries, and their combinations with “and”. The following inclusion criteria were used:—written in English,—full text article,—publication date: 1 January 1990–30 June 2023,—topic: exogenous risk factors for cardiovascular diseases. As a result of the search, 280 articles were collected (after removing duplicates and screening titles and abstracts). From them, 83 were included in the review (82 of them were published after the year 2000). Both original and reviewed articles were used to prepare the background for the review.

## 3. Results

### 3.1. Noise

Noise can be defined as audible sound energy that can affect human physical and mental health. It is believed, that exposure to noise above 85 dB is considered harmful, whereas exposure to over 140 dB can cause immediate damage [30]. Repeated overexposure to noise at or above 85 dBA can cause permanent hearing loss, tinnitus, and difficulty understanding speech in noise [31]. Noise-induced hearing loss (NIHL) is one of the most known consequences of chronic noise exposure, most often found among aircraft pilots, but there are also non-hearing-connected consequences for human health related to noise exposure. Various factors including duration of the exposure, frequency, and amplitude, are important for health problems related to noise exposure. Several occupations, such as aeronautical technicians, aircraft and helicopter pilots, and cabin crewmembers are connected with chronic exposure to large pressure amplitude of low-frequency noise (≥90 dB, ≤500 Hz, which includes infrasound ≤ 20 Hz) [32]. Effects on health and well-being which are caused by exposure to noise, with the exclusion of effects on the hearing organ and the effects which are due to the masking of auditory information (i.e., communication problems) are called non-auditory effects of noise [33]. Both acute and chronic noise exposure can be detrimental to mental and physical health. Low-frequency noise, which has been identified also as an environmental problem, is a powerful stressor involved in induction of emotional changes such as sleep disorders, agitation, annoyance, cognitive alterations, and dental wear but also high blood pressure and CVD development [34,35]. Noise pollution is associated with (i) general morbidity (illness), (ii) neuropsychological disturbances such as headaches, irritability, insomnia, fatigue, and neuroticism; (iii) cardiovascular system disturbances such as hypertension, hypotension, and cardiac disease; and (iv) digestive disorders such as ulcers and colitis, as well as endocrine and biochemical disorders [30].

Significant exposure–response dependance was found between night time aircraft noise exposure and risk of hypertension in people living around busy airports. Studies by Correia et al. confirmed a statistically significant association between exposure to aircraft noise and risk of hospitalization for CVDs among older people living in or near airport areas. Exposure to aircraft noise has been associated with physiological responses and psychological reactions, such as sleep disturbances, sleep disordered breathing, annoyance, and nervousness, and as well as hypertension outcomes, including increased blood pressure, self-reported hypertension, incidence of hypertension, and antihypertensive medication use [35]. Similar observations were obtained in numerous studies conducted all over the world [30]. For example, aircraft noise exposure around Schiphol Airport (Amsterdam, The Netherlands) has been related to more medical treatment for heart trouble and hypertension, more cardiovascular drug use, and higher blood pressure. Also, prevalence of hypertension was higher among people exposed to time-weighted energy-averaged aircraft noise levels of at least 55 dBA or maximum levels above 72 dBA around Arlanda airport (Stockholm, Sweden) [33].

There are several possible mechanisms of non-auditory effects of noise on human health, including detrimental influence of noise the on cardiovascular system. Noise causes oxidative stress, vascular dysfunction, autonomic imbalance, and metabolic abnormalities, further increasing the adverse health effects of classic risk factors such as arterial hypertension, diabetes, hypercholesterolemia, and smoking (e.g., accelerated progression of atherosclerosis and increased susceptibility to cardiovascular events). Munzel et al. proposed in their review paper the comprehensive mechanism of detrimental effect of aircraft noise. As a result, the sympathetic nervous system is over-activated and levels of noradrenaline, adrenaline, angiotensin II, and ultimately cortisol, the stress hormone, increase. Angiotensin II activates endothelial NADPH oxidase causing oxidative stress, which may induce direct scavenging of nitric oxide and endothelial nitric oxide synthase uncoupling through oxidation of tetrahydrobiopterin and endothelial nitric oxide synthase S-glutathionylation. Reactive oxygen species (ROS) link various pathways including PI3K/Akt-b1 signaling, FOXO transcription factors, TGF-β1 and NFκB signaling, as well as the ET-1 system, which increases circulating levels of IL-6 and plays an important role in the expression of vascular adhesion molecules. Superoxide and nitric oxide produced by infiltrating immune cells (neutrophils, natural killer cells, monocytes/macrophages) promote the formation of 3-nitrotyrosine, malondialdehyde, and 4-hydroxynonenal positive proteins, which further cause cell oxidative damage. Uncoupling of endothelial nitric oxide synthase not only decreases nitric oxide production, but also enhances existing oxidative stress. Endothelial nitric oxide production is further reduced by glucocorticoids such as cortisol, causing impaired vasodilation and increased blood pressure. Excess production of noradrenaline, adrenaline, and ET-1 enhances contraction, which is further potentiated by glucocorticoids. All these vascular changes contribute to the development of metabolic abnormalities assumed by elevated blood glucose levels [36].

Single unprotected exposure to impulse noise can produce hearing damage that is irreversible, whereas effects of steady-state noise manifest only after long-term repeated exposures [37]. Noise exposure can cause short-term physiological responses via the autonomic nervous system. It causes physiological activation such as increased heart rate and blood pressure, peripheral vasoconstriction, and increased peripheral vascular resistance. This is rapid habituation to brief noise exposure [33]. Prolonged noise exposure can also be considered a stressor. The relationship between noise exposure and the body’s stress response is critical because prolonged stress levels can increase the risk of life-threatening health conditions such as CVDs. The body’s stress response system attempts to maintain homeostasis through negative feedback of the hypothalamus–pituitary–adrenal (HPA) axis. The paraventricular nucleus of the hypothalamus activates the HPA axis through the secretion of adrenocorticotropin-releasing hormone (CRH), which triggers the release of adrenocorticotropic hormone (ACTH) from the anterior pituitary gland; ACTH then reaches the adrenal glands, which release stress hormones such as cortisol, adrenaline, and noradrenaline and activates their release. When levels of stress hormones become high enough, they interact with the hypothalamus and lumbar-cortex, severing the HPA axis, resulting in negative feedback [38].

Military personnel is exposed to loud intense sounds during both training missions and when deployed. This fact is well documented historically from World War II, as many military personnel returned from combat with impaired hearing [31]. Occupational noise exposures, including battlefield noise, may potentially exert serious consequences to hearing health, and also non-auditory effects on health including CVDs. Jokel et al. summarized noise sources in use, including military weapons, ground vehicles, airplanes, and ships, and confirmed that all military weapon systems exceed peak sound pressure levels of 140 dB, several classes of weapon systems expose crews to impulse noise levels exceeding 180 dBP, and almost all ground and air transport platforms expose crews and passengers to steady-state noise exceeding 85 dBA during operation [37].

Noise exposure in military aircrew resulting in NIHL is a well-known fact, which was confirmed in many studies. Impaired hearing of many veterans of World War II has led to the first recommended noise exposure limit in the U.S., which was promulgated by the U.S. Air Force in 1948. Updated regulations of the U.S. Air Force from 1956 specified the seven components of an effective hearing loss prevention program which are still recognized today: noise measurement, noise control, hearing protection, audiometric testing, training, record-keeping, and program evaluation [31]. Modern high-performance aircraft are more powerful and efficient, but also frequently generate high noise levels. More powerful and efficient engines frequently produce noise levels as high as 110 dB to 150 dB. Pilots are constantly exposed to aircraft noise, which can reach levels of 90 to 120 dB [39]. Rajguru et al. describe potential noise sources in various aircraft. In high-speed jet aircraft, noise is generated by external airflow around the fuselage canopy and forward fuselage structure, especially during aircraft maneuvers; and internally, noise is generated by airflow from air conditioning and cooling flows into the cockpit space and from cockpit air conditioning and pressurization systems. These jet aircraft noise levels are highest during low altitude flight (e.g., the overall noise level in the cockpit during a vertical landing is approximately 120 dB). Additionally, formation flying and aerial refueling increase noise levels. Helicopter aerodynamic noise is generated by the main and tail rotors, and, in the case of twin rotors, also includes the interaction between the rotors and the interaction between the rotors and the airframe. Mechanical noise in helicopters comes from rotating systems connected to the rotors, such as gearboxes, transmission shafts, transfer gears, auxiliary systems, and drive shafts. Noise in transport and reconnaissance aircraft comes primarily from propellers, rotors, wing-mounted gas turbines, boundary laminar flow, equipment cooling systems, and cockpit air conditioning systems [40]. The “quiet” spots are about 126 dBA. For a carrier launch, aircraft are hooked to the catapult and restrained, while the aircraft goes to full power for final aircraft checks. These noise levels are about 148 dBA for tactical aircraft and somewhat lower for propeller aircraft [37]. A study by Zevitas et al. also showed differences in the average noise levels observed throughout the different flight phases. The highest noise levels were detected during ascent and descent, where engine thrust (and consequently engine noise) would be greatest, and airframe noise due to airflow around lift and control surfaces such as flaps and landing gear would be greatest [41].

Many studies suggested that chronic exposure to noise in aircrew was a risk factor not only for noise-induced hearing loss but also non-auditory health effects. Exposure to environmental conditions and pollutants in the aircraft cabin may be associated with higher rates of some non-auditory effects observed in crew members compared to the general population including sleep disorders, depression, fatigue but also CVDs. One of the well documented non-auditory health effects is hypertension risk in pilots exposed to high noise levels. Studies by Siagian et al. in Indonesian Air Force pilots showed that high interior aircraft noise, high total flight hours, and high resting pulse rate, increased risk for high diastolic blood pressure. Pilots exposed to aircraft noise 90–95 dB had a 2.7-fold increased risk for high diastolic blood pressure in comparison to pilots exposed to noise of 70–80 dB (adjusted odds ratio (ORa) = 2.70; 95% confidence interval (CI) = 1.05–6.97). Increased diastolic pressure is the result of increased peripheral resistance, which in turn is mainly caused by a high sympathetic tone, and thus is reflected by a high resting pulse rate. Pilots with resting pulse rates of ≥81/min rather than ≤80/min had a 2.7-fold increase for high diastolic blood pressure (ORa = 2.66; 95% CI = 1.26–5.61). In terms of total flight hours, which are related to the duration of exposure to noise, pilots who had 1401–11,125 h rather than 147–1400 h had a 3.2-fold increase for high diastolic blood pressure (ORa = 3.18; 95% CI = 1.01–10.03) [39]. Also, studies by Tomei et al. confirmed that chronic exposure to noise is a risk factor for blood hypertension in pilots exposed to high noise levels, and that the drop in blood pressure may be a sign of more sensitive effect of noise on blood pressure. They observed that pilots who were at higher levels of exposure to noise were at higher risk of hypertension, particularly higher diastolic blood pressure [20]. Noise exposure also caused changes in arteries. Exposure to low-frequency noise appeared to cause arterial wall thickening and stiffness. Studies by Khoshdel et al., performed in male aeronautic technicians who were employed in a helicopter repair and maintenance plant and were exposed to wide band noise, showed no increase in carotid artery intima media thickness accompanied with an increase in arterial stiffness, which suggests another mechanism for wide band noise-induced arterial stiffness, and a significant decrease in serum cystatin C levels, which is known to be influenced by extracellular matrix metabolism [32].

### 3.2. Altitude and Hypoxia

The inherent physiological challenges of altitude include hypoxia (and hyperventilation), gas volume change, cold, and decompression sickness [21]. At high altitudes, a decrease in atmospheric pressure (hypobaria) causes a decrease in the partial pressure of inhaled oxygen (PiO_2_), resulting in hypoxic hypoxia. Hypoxia can be defined as a state of inadequate oxygen availability throughout the body caused by a decrease in the partial pressure of oxygen (PO_2_) in the atmosphere, a decrease in PiO_2_, and a loss of ventilation–perfusion equilibrium. A hypoxic cellular environment is caused by hypoxemia. Hypoxemia is a decrease in arterial partial pressure of oxygen (PaO_2_) and hemoglobin-bound oxygen saturation (SaO_2_), resulting in inadequate oxygen supply to the tissues. Acute hypoxia is a major physiological threat in military aviation during high-altitude flights and operational maneuvers. The detrimental effects of hypoxia in aviation were observed during World War II and established as a major limitation of military aircraft due to the lack of oxygen to the brains of aircrew in flight. Hypoxic hypoxia is caused by a decrease in the partial pressure of oxygen in the air. Exposure to hypoxic hypoxia causes an immediate physiological response in the respiratory, cardiovascular, cerebrovascular, and visual systems to maintain adequate tissue oxygen supply [42]. The brain is especially susceptible to oxygen deficit as it relies on oxidative energy metabolism, which is required to support neuronal signaling. Even minor impairments to the brain’s function resulting from hypoxemia, cerebral hypoxia, or hyperventilation-induced hypocapnia can be catastrophic in military aviation due to the dynamic and demanding environment within which the aircrew must operate [43]. Retina, similarly to the brain, also has the highest oxygen uptake per unit mass of any body system, which makes visual performance and sensitivity markers for hypoxia studies. Some visual consequences of hypoxia mentioned in the literature include a decrease in visual light sensitivity at 2255 m, and impairments in scotopic sensitivity, photopic and mesopic chromatic sensitivity, and photopic and mesopic contrast sensitivity [21].

Nowadays, hypoxia-related aviation fatalities are rare, but incidences of hypoxia are common, particularly in fighter and training aircraft. In aviation, altitudes up to 3048 m are regarded as a physiological zone where the impact of hypoxia on a pilot’s cognitive and psychomotor performance is relatively small and, therefore, has few implications on flight safety [44,45]. In most of the individuals exposed to altitudes above 3048 m (10,000 ft) signs and symptoms of hypoxia are common [43]. Hypoxia impairs a spectrum of cognitive domains (simple and choice reaction speed, processing speed, working memory, short-term memory, attention, executive function, and novel task learning). As summarized by Steinman et al., hypoxia has been shown to impair the working memory of pilots, influence their ability to process information, as well as impair complex decision making and increase reaction time. Studies performed using a flight simulator reported a significant decrease in flight performance at 4572 m—significantly more procedural errors being made, especially during descent and landing as well as higher variability in flight performance during hypoxia exposure of 5486 m [44]. Also, studies performed in helicopter pilots operating at high altitude revealed their decreased awareness of environment and alertness accompanied by the fact that the majority of pilots did not notice they were hypoxic or recognize their hypoxia symptoms during the simulation flight at 4572 m. This is of utmost importance due to the fact, that helicopter cabins are not pressurized and are often not equipped with oxygen systems [46]. Studies by Pighin et al. also revealed that mild hypoxia promotes riskier choices in the loss domain [47]. Studies by Bouak et al. indicated the influence of mild hypoxic hypoxia on fatigue and mood in aircraft pilots, showing that hypoxia significantly increased signs and symptoms as well as general and physical aspects of fatigue and decreased positive mood in more than 60% of the participants [48].

The severity of hypoxia can be based on the level of blood or tissue oxygenation, or hypoxic signs and symptomology. There is large inter-individual variation in hypoxia tolerance, which may, in part, be attributable to the magnitude of the hypoxic ventilatory response and cardiovascular reflex. According to a study by Sucipta et al., pilots meeting the physical exercise recommendation (>75 min/week of vigorous exercise) appeared to be at higher risk of having a useful awareness time of less than 4 min. This was consistent with the notion that physical exercise increases muscle mass and capillary density, thereby increasing the capacity for oxygen generation by muscles and, consequently, pilots with greater muscle mass are more prone to hypoxia [49].

Military pilots perform daily flying tasks, sometimes including emergency operations, as well as out-of-flight tasks, which contribute to varying levels of fatigue and cardiac vascular endurance, which can affect the decline in the pilots’ performance [49]. During military operations, while flying, or during physiological training these specific conditions can lead to hypoxia or risk from decompression illness, mountain sickness, pulmonary barotrauma, and cerebral arterial gas embolism. The influence of hypoxia on CVD risk in military pilots is not fully recognized. There are several studies, which may give small insight into this issue, but most of the results is indirect. Some data come from studies in athletes, who often use acclimatization at high altitudes to improve performance at sea level, as acute hypoxia triggers various cardiovascular changes that challenge the cardiovascular system, especially during exercise. Alterations in cardiovascular parameters: Heart rate and mean blood pressure were apparent only after exercise load occurred at approximately 3000 m. Short-term exposure to hypoxia in a hypobaric chamber at 4000 to 5000 m induced acclimatization to altitude and improved aerobic endurance affecting mainly the adaptive respiratory and circulatory responses [42]. Enhanced sympathetic tone and pulmonary artery vasoconstriction occurs during acute hypoxia. The cardiovascular system increases cardiac output to meet the metabolic needs of the working muscle during acute hypoxia. Mulliri et al. detected significant reductions in peripheral arterial oxygen saturation and cerebral oxygenation but none of them were changed in response to exercise in hypoxia in comparison with exercise in normoxia. They concluded that the circulatory system well tolerates this particular type of hypoxic exercise, which does not pose any significant challenge for the cardiovascular system [50]. Alterations in cardiovascular parameters: Increased blood pressure and heart rate, as well as acute reduction in cerebral regional oxygen saturation were found to occur only after exercise load at approximately 3000 m altitude in unacclimatized subjects [42]. It seems important to take into account that tasks of military pilots during flight are considered as a type of physical activity, not as static work, which allows this small analogy.

Very interesting are results regarding CVD risk connected with living at moderate and high altitude. A review by Savla et al. presented this dependance. Changes in the cardiovascular system observed in hypoxia include reductions in the left ventricular and diastolic volume, increased levels of coronary blood flow to accommodate the reduction in arterial oxygen content, sympathetic hyperactivity leading to downregulation of beta receptors, changes in hemoglobin–oxygen affinity, and hypoxia-mediated erythropoietin production, which stimulates red blood cell production. Presented studies show a reduced risk of CVDs with increases in altitude. Epidemiological data revealed a lower incidence of heart diseases in high-altitude states and elevated high density lipoprotein cholesterol (HDL-C) levels, as well as a negative association of dyslipidemia, diabetes, insulin resistance, obesity, metabolic syndrome, and elevated C-reactive protein levels with living at high altitudes. However, they also mentioned an increased risk of hypertension in high-altitude populations from China, Japan, Tibet, and Peru [51]. A narrative review of Aragon-Vela revealed lower than average CVD mortality in flight crews, whereas miners and soldiers exposed to intermittent hypoxia experienced increased risks of CVDs, which may result from socioeconomic status and lifestyle factors [52].

Changes in the oxygen partial pressure in the arterial blood perfusing the brain and the peripheral chemoreceptors affect the heart through autonomic nervous control. These indirect effects of hypoxia are prevalent and determine the response of cardiovascular parameters. Acute hypoxia modulates the sympathetic and parasympathetic cardiac activity. While studies in normal subjects did not show increased sympathetic stimulation with parasympathetic blockade or effects of hypoxia on autonomic cardiac control, co-activation of activity in both autonomic branches was observed in animal studies. A decrease in the partial pressure of oxygen (PaO2) of arterial blood stimulates peripheral chemoreceptors in the carotid and aortic arteries. As a result, the frequency of afferent nerve fiber impulses from the chemoreceptors increases, stimulating the vasoconstrictor area. In parallel with the activation of the chemoreflex, changes in PaO2 regulate ventilation, vascular resistance, arterial pressure, heart rate, cardiac output, and myocardial contractility via autonomic functions. Studies of pilots and crew members have shown that a reflexive cardio-vascular response to hypoxia causes an increased heart rate and a moderate increase in systolic blood pressure. Tachycardia, moderate increases in systolic blood pressure of 3 to 18 mmHg, and decreases or increases in diastolic blood pressure of 10 mmHg or less were observed in military pilots exposed to hypoxia at 5000 m elevation, but in most cases, hypoxia had no effect on systolic or diastolic blood pressure. Pilots did not exhibit symptoms of cardiac or vascular precipitation or collapse after hypoxia exposure, confirming that they were tolerant to hypoxia. The authors also observed a pattern of increased sympathetic and parasympathetic activity, with a relative predominance of vagal over sympathetic cardiac activity in the control of heart rate variability, as evidenced by a significant increase in mean R–R interval values and a slight indirect downward trend in sympathetic cardiac activity and diastolic pressure values [42].

### 3.3. Acceleration Forces

Acceleration forces (also called G-forces) are stressors, along with altitude, that are unique to fighter pilots and may be experienced by pilots during flight. Significant fluctuations in G-forces are not normally experienced by humans, but fighter pilots and high-performance aerobatic pilots experience them during aerobatic flight training, competitions, and air show maneuvers [53]. Maneuvers pulled by pilots may cause them to experience acceleration forces as high as 9 G. Positive G-forces (+Gz) are experienced when pulling into a climb, out of a dive, or into an inside loop and they result in the feeling of being pressed down into the seat. Then, the blood flow toward the brain becomes more difficult. Positive G-forces have been found to be associated with musculoskeletal symptoms and spinal shrinkage. Negative G-forces (-Gz) are experienced when pushing over into a dive or entering an outside loop and they result in the feeling of being lifted out of the seat. Then, the blood flow away from the brain becomes more difficult. Negative G-forces have been associated with bradycardia. A microgravity state has been connected with several adverse effects including intravascular contracture, decreased oxygen-carrying capacity, a decrease in heart rate, and abnormal sympathetic autonomic responses [53,54]. To the best knowledge of the authors, no direct influence of G-forces has been investigated in relation to CVD risk in pilots. However, the observation that membrane (both lipid and cell) fluidity is gravity-dependent (increased gravity results in decreased fluidity) may implicate the possible effect of G-forces on cardiovascular conditions. With regard to CVD risk, a study by Wochyński et al. should also be mentioned. They observed a positive correlation between total time exposed to positive G-force and HDL and Apo-A1 concentrations and an inverse correlation with blood serum TAG. They estimate that the significant increase in HDL levels is associated with the increase in testosterone levels observed after 10 weeks of training in first-year students at the Polish Air Force Academy in Dęblin [55].

### 3.4. Stress

Stress can be defined as mental or emotional strain and tension resulting from adverse or demanding circumstances, which creates physical and psychological or emotional imbalances within a person. Stress is caused by stressors, which can be considered as any activity, event, or other stimulus that causes stress [56]. The main physiological response of an organism to stress involves the HPA axis and the autonomic nervous system (ANS), composed of the parasympathetic nervous system (PNS) and the sympathetic nervous system (SNS) [57]. The HPA axis is the primary stress response acting through cortisol, which has been also involved in CVD risk. Hyperactivity of the sympathetic adrenal system (SAS) resulting in dysregulation of adrenaline and noradrenaline secretion in response to stressors, may exert detrimental effects on blood pressure, cardiac function, endothelial functioning, platelets, and metabolic function. Both the PNS and the SNS exert their effects on the cardiovascular system manifesting in regulation of heart rate variability. A low heart rate reflects hyper- or hypoactivity of these two parts of the ANS and is also connected with CVD risk. All of these confirm that stress is one of the important CVD risk factors [58]. Traditional risk factors do not fully explain the CVD risk and there is increasing awareness of the impact social, environmental, and psychological factors have on CVD incidence and outcomes. Emotional states as well as acute and chronic stressors have been linked to the pathogenesis of CVDs. A review by Neylon et al. presents the main evidence of psychosocial risk factors for CVDs. Permanent stress was found to be significantly and independently associated with non-fatal myocardial infraction or CVD death, psychosocial stress was associated with an increased risk for all types of stroke, patients who reported high levels of stress had double the risk of fatal stroke, and men with the highest self-reported stress levels were seen to have a two-fold increase in stroke—these are only a few examples confirming the great meaning of stress to cardiovascular health [59].

Occupational stress (or work-related stress) is the negative response people have to excessive pressures or other types of demands placed on them at work [56,60]. Each position is characterized by its own demands and stressors, but there are some occupations which are considered much more stressful. Both aviation and military are among them. As revealed by Urazaliyeva et al., emotional stress and bad behavioral habits are the most detrimental factors influencing the health of military personnel [61]. The aviation environment is rich in potential stressors such as hypoxia, acceleration, temperature, noise and communication, decompression sickness, exhaust fumes, vibration, and motion sickness. Pilots, like other remote and shift workers, experience many physical, emotional, and environmental stressors, including disturbed sleeping patterns, fatigue, sedentary work, anti-social work hours, isolation, long hours, high task loads, etc. They are subjected to different amounts of stress during all phases of flight [56,60,62]. Work-related stress has been linked with CVD risk [59]. Experiencing severe stress during traumatic events may lead to posttraumatic stress disorder (PTSD), which is a prevalent, chronic, and often disabling state, characterized by re-experiencing, avoidance, and hyperarousal symptoms. One of the most severe stressors in human life is war, and the prevalence of PTSD among active duty military personnel and war veterans is particularly high. General risk of CVDs has been greater in American veterans. PTSD has been also associated with increased risk for coronary artery disease and related mortality. The incidence of arterial hypertension, cardiac arrhythmias, and myocardial infarction was significantly higher in Croatian war veterans suffering from PTSD [63]. Many studies confirmed the association of PTSD and higher risk of CVDs and several mechanisms have been proposed: (i) PTSD is associated with stress-coping behaviors (tobacco, alcohol dependence, binge drinking, lack of exercise, etc.), which are also related to health and directly affect health status, including CVD risk; (ii) PTSD has direct negative effects on the central nervous system (CNS) (it alters the brain structure and function and the neurological perception of stimuli); (iii) stress reactions cause immune dysregulation and contribute to the development of CVDs. Many studies have confirmed the prevalence of metabolic syndrome (obesity, hypertension, unhealthy lipid profile—elevated total cholesterol levels, elevated low-density lipoprotein cholesterol (LDL), diminished high-density lipoprotein cholesterol (HDL), elevated triglyceride levels (TAG), elevated homocysteine levels), which are associated with an increased risk of cardiovascular mortality, that have been observed in patients with PTSD. The high prevalence of hypertension in patients with PTSD may be due to PNS overdrive, which may be associated with long-term changes in receptors that regulate blood pressure. The results of this study suggest that smoking may be associated with a higher prevalence of hypertension in PTSD patients. Smoking may be used as a coping strategy against trauma memories and negative emotions, and is more common in individuals with pre-existing PTSD. Higher anxiety sensitivity is associated with lower motivation to exercise in PTSD patients; PTSD is also characterized by ANS hyperactivity. Negative emotions chronically overstimulate the ANS, resulting in exaggerated cardiovascular reactivity to stress. Traumatic stress stimulates the CNS and hypothalamic pituitary tract, resulting in allostatic overload, platelet activation, inflammation, arterial hypertension, central obesity, insulin resistance, and endocrine dysfunction. PTSD is also characterized by a prolonged cardiovascular response to stress. Stress also increases the release of inflammatory cytokines involved in platelet activation and endothelial dysfunction, which in turn contribute to the development of atherosclerosis, one of the risk factors for CVDs [58,63,64,65]. Sleep disturbance may also play an important role in the development of CVDs. Studies by Ulmer et al. performed in American military personnel confirmed a relationship between sleep disturbances, PTSD, and depression as well as smoking and elevated blood pressure [66]. This finding may be of interest regarding pilots and the specificity of their work.

Studies by Mesko et al. performed in Slovenian military pilots confirmed that specific personality characteristics are common in pilots. These traits include: emotional stability, extraversion, sociability, conscientiousness, balance, and orientation towards actions and activity. They also revealed that military pilots who experienced many stressful events or stressors preferred specific ways of coping with stress. Cognitive avoidance and emotional discharge or emptying were significantly different in the stress-coping strategies of military pilots than in the control group. Pilots rarely look within themselves (introspection), blame each other, fight, or act childishly. They tend to eliminate emotions and cope more effectively with stress when they are faced with a problem or when their coping strategies are problem-centered and require immediate response or action [67]. In a study of Canadian military pilots, Hohmann et al. also found that when stressed, pilots reported just focusing on the task and doing whatever it takes to achieve their goals. However, pilots recognized the importance of stress management as a critical aspect of successful performance in all aspects of flight. Pilots reported many stress-coping strategies, including exercise, positive self-talk, meditation, vacations, compartmentalization, and socializing [68].

### 3.5. Tobacco Smoking

In general, tobacco smoking shortens an individual’s life expectancy by an average of 10 years, and each cigarette shortens one’s life expectancy by 15 min [69]. Tobacco smoking is the leading cause of preventable death in the general population of the United States, responsible for 20% of preventable deaths annually [70,71]. It also causes 30% of all heart diseases [72].

The health effects of tobacco smoking include CVDs, respiratory diseases, and cancer, as well as short-term adverse effects, such as acute respiratory illnesses, impaired wound healing, periodontal disease, and peptic ulcer disease. Tobacco smoking is one of the leading risk factors for CVDs, with an extreme impact not only on health but also on general wellbeing or work ability. It increases one’s vulnerability to injury on the job, increases the risk of mental health problems, and increases the risk of productivity loss [69,73]. Smoking is an important risk factor not only for CVDs, but also for cancer, and other chronic noncommunicable diseases. Mechanisms of smoking influence on CVD risk include a direct association of smoking with body oxidative stress through the generation of ROS, which in turn lead to endothelial dysfunction, which is an early event in atherogenesis. Smoking also oxidizes both HDL and LDL. Oxidized HDL loses its protective functions and may become atherogenic because of the reverse cholesterol transport. Smoking also elevates plasma fibrinogen levels and inflammatory marker levels (C-reactive protein (CRP) and interleukin-6 (IL-6)), which indicates an increase in the inflammatory process and changes in the thrombosis/fibrinolysis system. Smoking is also connected with hypercholesterolemia appearance, which is also a risk factor for CVDs [74] as well as with a decrease in the HDL level, an increase in blood pressure, an increase the blood’s tendency to clotting, and a decrease in physical performance. The risk caused by tobacco smoking increases when associated with other CVD risk factors [9,72].

Smoking is also a common health problem faced by military personnel all over the world (e.g., in Uganda, the U.S.A., and Taiwan). Studies have demonstrated a high prevalence of atherosclerosis and CVDs among armed forces personnel killed in combat or from unintentional injuries in Korea, Vietnam, Afghanistan, and Iraq [72]. Military service has been recognized as a risk factor for tobacco use [75] and in the military the smoking prevalence is higher than in the general population. In the U.S.A. in 2008, about 33% of soldiers in the Army smoked cigarettes [71]. In 2011, cigarette use in the military in the U.S.A. was also higher than in the civilian population (24.0% vs. 21.2%) [76]. Similarly, in Taiwan about 30% of military personnel had a smoking habit (Au-Yeung et al., 2021 [69]). Studies by Basaza et al. showed that 35.7% of male and 25.0% of female soldiers in Uganda smoked cigarettes [75]. These odds were significantly higher than those found in the general populations. In Lithuania, the rate of current smokers of military personnel among men was 2.6 times higher than that among women (45.9% vs. 17.9%; *p* < 0.001) [74]. Smoking rates have been consistently higher than in civilian populations not only among U.S. Service members but also among veterans, as approximately 30% of veterans reported tobacco use in the 2010–2015 National Survey on Drug Use and Health. Service members tend to smoke more while approaching their separation from the military. The high prevalence of smoking among veterans continues to have significant health, social, and economical effects (e.g., smoking-related health care costs are estimated by the Department of Veterans Affairs at above USD 2 billion annually) [73,77]. Military personnel in the U.S.A. were also more than twice as likely as civilians to use smokeless tobacco (use moist snuff or chewing tobacco) but there were differences among different branches of military, e.g., Army personnel were heavier users of tobacco and might have been more addicted to nicotine than Air Force personnel [78]. Electronic cigarettes (e-cigarettes) are another method of nicotine intake and typically include a diverse group of devices that allow users to inhale an aerosol containing nicotine, flavoring, and other additives. E-cigarettes typically contain fewer harmful substances than traditional combustible tobacco smoke and have been found to be used by approximately 14.4% of military personnel in the U.S. military. Tobacco smokers used e-cigarettes to quit smoking or reduce tobacco use, whereas nonsmokers used e-cigarettes for taste and flavor. An increased likelihood of using e-cigarettes was significantly and independently associated with younger age, lower officer rank, lower perceived harm of use, and use of other tobacco products. In general, the majority of participants believed that e-cigarettes were less harmful than cigarettes, but information on the long-term health consequences of e-cigarette use was lacking [79].

Factors that lead military personnel to use tobacco include ways to reduce or manage stress, anxiety, and boredom, and to cope with anger and other negative emotions. Soldiers working in stressful environments are more prone to PTSD, which increases the likelihood and risk of smoking. Exposure to deployment and combat is also associated with increased initiation and recidivism of cigarette smoking and use of smokeless tobacco. Historically, smoking has also been part of military culture (camaraderie, social norms, peer pressure) and has not only been accepted but even encouraged. Other historical reasons for tobacco smoking as part of the military culture and the long story of tobacco use in the U.S.A. were the inclusion of cigarettes in military rations from World War I to 1975, the increased availability of low-cost tobacco products at military installations, a long history of tobacco company sponsorship for military recreational activities, and tobacco advertising directed at young adult males. Other risk factors for smoking include lower educational attainment, younger age, having close friends who smoke, and previously being deployed abroad [73,75,76,79].

Smoking is known as negative marker for military performance, compromises physical fitness, increases absenteeism and presenteeism, and decreases productivity caused, i.a., by additional non-sanctioned breaks during the work day. It affects military readiness by reducing physical ability and endurance, decreasing alertness and cognitive function, increasing the risk of motor vehicle accidents and other unforeseen injuries, and resulting in absenteeism. It also places an extreme burden on the military in terms of physical readiness, long-term medical costs, and increased risk of early discharge. Smoking causes physical problems both individually and cumulatively because of its tremendous impact on national security risks, the nation’s military strength, and combat readiness [69,71,73,75].

Effects of smoking are of particular concern to military aviation where optimum physiological functioning is critical. Pilots need to cope with the potentially harmful effects of altitude, occupational toxins, environmental hazards, mission requirements, and enemy activity. Both nicotine and carbon monoxide (CO) exposure are of special importance regarding military pilots. Nicotine is a potent psychoactive compound, which stimulates the CNS, and a ganglionic compound, which improves information processing and enhance sensorimotor performance, which was confirmed by increased cognition and coordination of pilots using nicotine while performing critical tasks in a flight simulator. Nicotine also induces sympathetic neural stimulation which causes increased blood pressure and increased heart rate, which can be considered protective to sustaining G-forces. However, prolonged impact of nicotine is detrimental to the cardiovascular system. On the other hand, due to these stimulatory properties, nicotine should be avoided before bedtime as a potential disruptor of restorative sleep, increasing fatigue and the chance for inflight errors. Absence of nicotine induces withdrawal symptoms (anger, anxiety, depression, difficulty concentrating, impatience, insomnia, irritability, and restlessness) and distress, which also can be dangerous as it causes the loss of sustained vigilance and task management, resulting in increased error rate during flight and breakdown of crew coordination [70]. CO present in tobacco smoke exerts poisonous effects as it decreases resistance to hypoxia at altitude (due to binding to hemoglobin and competitive inhibition of cytochrome c oxidase) and degrades visual performance causing degradation of night vision, decreased contrast sensitivity, and longer dark adaptation [70]. Studies by Kotamiiki et al. on male military pilot candidates revealed the influence of smoking on blood pressure (significantly higher diastolic blood pressure in smokers in comparison to non-smokers, especially in stressful situations). Smokers also had higher levels of noradrenaline and renin both at rest and after exercise, implying increased activity in sympathetic function and the renin–angiotensin system, as well as enhanced excretion of thromboxane, which results in increased vasoconstrictive properties of smoking [80]. The ability to acutely increase sympathetic activities and plasma catecholamine levels has been recognized as a major mechanism for increased risk of coronary artery disease in smokers. Chronic smoking also results in the elevated plasma noradrenaline levels seen in elderly subjects and tends to increase the resting heart rate and decrease heart rate variability [80]. Studies by Vaicaitiene et al. in Lithuanian military personnel revealed that cigarette smokers had higher total cholesterol levels than non-smokers did [74]. According to American studies, Army (AOR 1.52, 95% CI [1.34, 1.72]), Navy (AOR 1.36, 95% CI [1.21, 1.53]), Marine Corps (AOR 1.28, 95% CI [1.14, 1.43]), and Coast Guard (AOR 1.22, 95% CI [1.07, 1.20]) personnel were more likely than Air Force personnel to smoke tobacco [76]. Other American data claim that the Army has the second highest smoking rate (26.8%) behind the Marine Corps (30.7%), followed by the Navy (24.3%), Coast Guard (19.9%), and Air Force with the lowest smoking rate (16.7%) [72].

### 3.6. Oral Health

Periodontal diseases are infections that are caused by microorganisms that colonize the tooth surface at or below the gingival margin and which can manifest with different clinical presentation. They are considered to be multifactorial diseases developing as a result of complex interactions between the patient and the environment with a great impact of bacteriologic, immunologic, and genetic factors involved in their pathogenesis. As inflammatory pathologies, they are connected with the production of inflammatory mediators such as tumor necrosis factor α (TNF-α), interleukins (IL-1, IL-6, IL-8), and C-reactive protein (CRP). Periodontal diseases are also considered among CVD risk factors, especially involved in atherogenesis development. Bacteria of the oral flora such as *Porphyromonas gingivalis*, *Aggregatibacter actinomycetemcomitans*, and *Streptococcus sanguis* have been found in patches of atheroma, probably due to transient bacteremia by crossing the dental sulcus. Moreover, periodontal disease and CVDs share some of the risk factors, e.g., tobacco smoking and stress, both of them of great concern in military personnel, especially in pilots and crew members. Studies by Thankappan et al. revealed the prevalence periodontitis to be 7.9% of the total examined population of military aviators. In those actively involved in flying the incidence rate was 10.3% whereas in ground duty branches—5.6%. Among pilots, 12.2% of helicopter crews, 10.3% of fighter streams, and 8.9% of the transport crews suffered from periodontitis. The study also revealed that tremendous stress suffered by the military flying crew when performing duty, affected the periodontal health [81]. Studies by Bell Ngan et al. performed in Cameroonian soldiers revealed the prevalence of periodontal diseases to be 79.0% (the frequency of gingivitis was 64.4% and the frequency of periodontitis was 15.6%, respectively) in the examined population. They also confirmed tobacco smoking being a common risk factor for periodontitis and CVDs (odds ratio: 4.44 [1.73–11.43], *p* = 0.0031), and indicated that poor oral hygiene, which is a well-known risk factor for periodontitis, also increases the risk of CVDs [82].

One of the factors causing periodontal deterioration, temporomandibular joint disorders, tooth wear, chipping of teeth and prostheses, masticatory muscle pain, and sensitive teeth, is bruxism, which is characterized by clenching or grinding of teeth or bracing or thrusting of the mandible, while awake or asleep, either voluntarily or involuntarily. The close relationship between anxiety and depression, as well as stress with bruxism has been established. Regarding military pilots, occupational stress is of great concern. Al-Khalifa established in Saudi Arabian fighter pilots that they experienced higher occupational stress than non-pilots (45.5% vs. 27.3%) and that the prevalence of bruxism was higher among pilots than in non-pilots (52.7% vs. 30.9%), which made them 3.9 times more likely to have the combination of those two pathological conditions [83]. This may indicate that there is a direct cause end effect relationship between occupational stress, oral stress, and CVD risk.

## 4. Conclusions

The amount of evidence confirms that CVD risk may be diminished by elimination or limitation of modifiable risk factors. Some of them are specific occupational and environmental risk factors related to specific work conditions. Both military professionals and military aviators are exposed to certain characteristic conditions (Table 1), which may influence the CVD risk, but the number of solid scientific data regarding this subject are limited. There is a great need to perform complex studies on environmental and occupational risk factors for CVDs in military personnel and military aviators as well as how to minimize their influence, to prolong the state of health and military suitability of this professional groups.

## Figures and Tables

**Figure 1 jcm-12-07492-f001:**
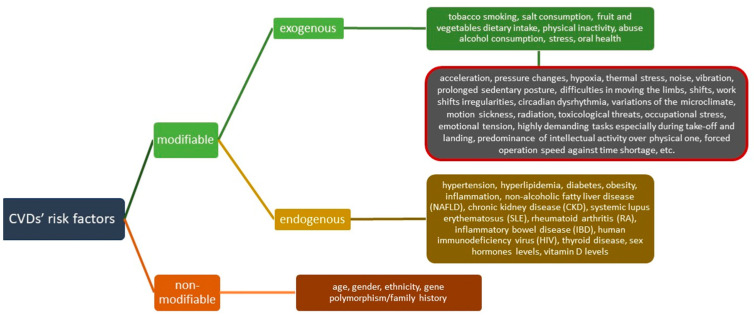
CVD risk factors with a special emphasis on environmental and occupational factors typical for military professionals and military pilots and crew members. CVDs’ (cardiovascular diseases).

**Table 1 jcm-12-07492-t001:** Summary of exogenous risk factors’ influence on CVD risk.

Risk Factor	Cardiovascular Implications	References
Noise	Increased susceptibility to cardiovascular events, cardiac disease, hypertension, higher diastolic blood pressure, hypotension, oxidative stress, vascular dysfunction, impaired vasodilation, peripheral vasoconstriction, increased peripheral vascular resistance, arterial wall thickening and stiffness, accelerated progression of atherosclerosis, elevated blood glucose levels, increased heart rate and blood pressure, high resting pulse rate	[20,30,32,33,35,39]
Altitude and hypoxia	Lower level of blood or tissue oxygenation, pulmonary barotrauma and cerebral arterial gas embolism, alterations in heart rate and mean blood pressure, enhanced sympathetic tone and pulmonary artery vasoconstriction, reductions in peripheral arterial oxygen saturation and cerebral oxygenation, increased blood pressure and heart rate, acute reduction in cerebral regional oxygen saturation, elevated HDL levels, increased risk of hypertension, changes in the oxygen partial pressure in the arterial blood perfusing the brain, modulation of the sympathetic and parasympathetic cardiac activity, decrease in the partial pressure of oxygen of arterial blood, stimulation of peripheral chemoreceptors in the carotid and aortic arteries, regulation of vascular resistance, arterial pressure, heart rate, cardiac output, and myocardial contractility, increased heart rate and systolic blood pressure, tachycardia, decreases or increases in diastolic blood pressure, significant increase in mean R–R interval values, and a slight indirect downward trend in sympathetic cardiac activity and diastolic pressure values	[42,49,50,51]
Acceleration forces	Intravascular contracture, decreased oxygen-carrying capacity, decrease in heart rate, abnormal sympathetic autonomic responses, modulation of membrane fluidity, increased levels of blood serum TAG, increase in HDL levels	[53,54,55]
Stress	Effects on blood pressure, effects on cardiac function, influence on endothelial functioning, platelets, and metabolic function, regulation of heart rate variability, low heart rate, non-fatal myocardial infarction, CVD death, increased risk for all types of stroke including fatal stroke, greater general risk of CVD, increased risk of coronary artery disease and related mortality, higher incidence of arterial hypertension, cardiac arrhythmias and myocardial infarction, hypertension, elevated total cholesterol levels, elevated LDL, decreased HDL, elevated TAG levels, elevated homocysteine levels, changes in receptors that regulate blood pressure, allostatic overload, platelet activation, inflammation, arterial hypertension, central obesity, insulin resistance, increased release of inflammatory cytokines involved in platelet activation and endothelial dysfunction, elevated blood pressure	[58,59,63,64,65,66]
Tobacco smoking	Increased oxidative stress through generation of ROS, endothelial dysfunction, oxidation of both HDL and LDL, reverse cholesterol transport, elevated plasma fibrinogen levels and inflammatory marker levels (C-reactive protein (CRP) and interleukin-6 (IL-6), increase in the inflammatory process, changes in the thrombosis/fibrinolysis system, hypercholesterolemia, decreased HDL levels, increased blood pressure, increased blood tendency to clotting, increased heart rate, enhanced excretion of thromboxane, which results in increased vasoconstrictive properties of smoking, increased sympathetic activities and plasma catecholamine levels, elevated plasma noradrenaline levels, increased resting heart rate and decreased heart rate variability, higher total cholesterol levels	[9,70,72,74,80]
Oral health	Production of inflammatory mediators such as tumor necrosis factor α (TNF-α), interleukins (IL-1, IL-6, IL-8), and CRP, presence of *Porphyromonas gingivalis*, *Aggregatibacter actinomycetemcomitans*, and *Streptococcus sanguis* in patches of atheroma	[82]

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
