# Peer review of "Selected Exogenous (Occupational and Environmental) Risk Factors for Cardiovascular Diseases in Military and Aviation"

_jcm, 2023, doi:10.3390/jcm12237492_

Round 1

Reviewer 1 Report

Comments and Suggestions for Authors

I really appreciated reading this review article. It is well-written, well structured and easy to follow. The topics discussed are very interesting, but all related to a very specific sector: the health applied to the military field. In my opinion this is a weakness of the manuscript, because the article will not be read by as many scientists. Moreover, this Journal is open towards many different disciplines and the readers are not only soldiers. For this reason I suggest improving much more the introduction (or, better, creating a new paragraph) in which main molecular basis of these diseases are reported or mentioned. This would attract the attention of more scientists. For example, there are so many connections in basic science between the items mentioned in the text (i.e., hypoxia, altitude, stressors, tobacco, inflammation, ect) and mitochondria, organelles whose function and dysfunction is the main hallmark of CVDs. Here I report some papers that could be mentioned in the new version of the review, and that suggest these existing links to the authors (PMID: 35205167, 37446282, 35131483).

Also, a figure may help to summarize the concepts of this review.

Author Response

Reviewer 1

Journal JCM (ISSN 2077-0383)

Manuscript ID jcm-2655141

Type Review

Title: Selected exogenous (occupational and environmental) risk factors of cardiovascular diseases in military and aviation

Authors: Ewelina Maculewicz * , Agata Pabin , Łukasz Dziuda , Małgorzata Białek , Agnieszka Białek *

I really appreciated reading this review article. It is well-written, well structured and easy to follow. The topics discussed are very interesting, but all related to a very specific sector: the health applied to the military field. In my opinion this is a weakness of the manuscript, because the article will not be read by as many scientists. Moreover, this Journal is open towards many different disciplines and the readers are not only soldiers. For this reason I suggest improving much more the introduction (or, better, creating a new paragraph) in which main molecular basis of these diseases are reported or mentioned. This would attract the attention of more scientists. For example, there are so many connections in basic science between the items mentioned in the text (i.e., hypoxia, altitude, stressors, tobacco, inflammation, ect) and mitochondria, organelles whose function and dysfunction is the main hallmark of CVDs. Here I report some papers that could be mentioned in the new version of the review, and that suggest these existing links to the authors (PMID: 35205167, 37446282, 35131483).

We would like to thank Reviewer 1 for their time and effort spent in reviewing our manuscript. All the remarks were very useful and inspiring, and they helped us to improve our manuscript. All the remarks were addressed below and proper modifications have been introduced into manuscript’s body (red font). We really hope that Reviewer 1 will find them suitable and sufficient.

Regarding the remark of possible limited interest of our paper to the JCM readers we would like to explain, that this review has been prepared as a part of project funded by Ministry of Health in 2021–2025 as part of the National Health Program (Grant/Award Number: 364/2021/DA of 29 November 2021), which focuses on military personnel only. However, we would like to emphasize that some of the exogenous factors, reviewed in this manuscript, such as hypoxia, altitude, G-forces or even tobacco smoking or oral health, are shared also by the civilian aviators. Our aim was to draw the attention of general audience to the meaning of exogenous risk factors of CVDs, which are less investigated than endogenous. Military pilots and crew members seem good model for such investigations due to the good characteristic not only of the population but also of many co-occurring factors. We would like to mention also, that the first part of our studies, regarding endogenous risk factors of CVDs in military pilots, has been published in JCM in 2022 and has been viewed 1376 times (cited 1 time so far), which is slightly higher score than for other papers published in the same Special Issue of JCM. It makes us optimistic regarding the interest of JCM Readers.

 We really appreciate this contribution, especially the enhancement to include to ‘introduction’ section an additional passage regarding the molecular basis of CVDs and the meaning of mitochondria in CVDs development, as well as some suggestion concerning the papers. We tried to make this additional passage short whether informative and we really hope that the Reviewer 1 will find it sufficient.

Some of the factors (e.g. tobacco smoking, hypoxia, altitude, stress and different stressors) share the same pathways of the contribution to CVDs development and progression, including oxidative stress and inflammation being of special meaning for atherosclerosis, which is considered as an intermediate in CVDs development. It begins at sites of endothelial injury and consists of three stages: initiation (encompassing endothelial injury and disfunction, atherogenic lipid deposition and proinflammatory state), progression (development of atherosclerotic plaques) and complications (limitation of myocardial blood flow or plaques rupture, which triggers clot formation and leads to the acute obstruction of the arterial lumen resulting in clinical events).

Oxidative stress and inflammation are interrelated and they form a vicious cycle during atherosclerosis and CVDs development and progression. Oxidative stress leads to the oxidation of LDL. The oxidative forms of LDL-C (oxLDL) are more easily taken up by macrophages and are involved in the formation of foam cells. The lipid-laden macrophages are deposited underneath the endothelium of arteries taking part in atherosclerotic plaques formation. Reactive Oxygen Species (ROS) also initiate expression of adhesion molecules, stimulation of vascular smooth muscle proliferation and migration, apoptosis in the endothelium, the oxidation of lipids, the activation of matrix metalloproteinases, and the alteration of vasomotor activity. It is of special importance in case of military pilots and crew members as flying conditions (like high altitude and hypoxia) decrease oxygen availability and increase the formation of ROS. Oxidative stress activates transcription factors that alter inflammatory cytokines, soluble mediators, and chemokines. Hence, cytokines released by inflammatory cells gather inflammatory cells to the sites of inflammation, leading to increased ROS production, which confirms strong dependence between oxidative stress and inflammation. Vascular inflammation contributes to atherosclerosis by numerous mechanisms including: an increased expression of adhesion molecules and their ligands, an extravasation of leukocytes, an activation of pro-inflammatory signaling pathways, and an increase in oxidative stress as well as cytokine production in the arteries. Detailed interplay of oxidative stress and inflammation has been previously described (Maculewicz et al., 2022; Molavi & Mehta, 2004; Nayor et al., 2021).

It should be mentioned the importance of mitochondria as a crossroad where oxidative stress and inflammation are bended in CVDs progression. Mitochondrial oxidative stress is involved in endothelial disfunction and apoptosis of cardiomyocytes. Also dysregulation of antioxidant defense system, especially disruption of antioxidant enzymes activity, contributes to mitochondrial dysfunction and is strictly involved in CVDs progression. ROS production is also related to mitochondria disfunctions as they destroy the electron transport chain, leading to further ROS formation, aggravation of mitochondrial dysfunction and cytochrome c release, and aggravation of the intrinsic apoptotic pathway in cardiomyocytes. Hypoxic conditions contributes to the development of CVDs also by the fact, that mitochondria sense the diminished level of oxygen and are considered as the main site of hypoxic damage. However, the influence of hypoxia acting via hypoxic-inducible factor-1 (HIF-1) is dual. Acute hypoxia leads to accumulation of HIF-1, its transfer to nucleus and activation of genes responsible for shift of aerobic ATP production with electron transport chain to glycolysis which is Ca2+ upregulated with consequent acidification of the environment, sodium and overload and breakdown of ATP production. In this condition HIF-1 exerts cardioprotective effect by alleviating ROS generation and stimulating the removal of unwanted mitochondria through mitophagy. In chronic hypoxic conditions HIF-1 increases ROS levels, stimulates the inflammasome genes expression (including interleukin 1ß (IL1ß) gene) and contributes to atherosclerosis development, which in turn confirms the importance of hypoxia in CVDs development (Bouhamida et al., 2022, 2023; Liang et al., 2022).  

Also, a figure may help to summarize the concepts of this review.

As suggested by the Reviewer 1, a figure has been included.

Reviewer 2 Report

Comments and Suggestions for Authors

Dear Editor and Authors,

I read the review paper entitled "Selected exogenous (occupational and environmental) risk factors of cardiovascular diseases in military and aviation". The topic is original and manuscript discuss the influence of selected occupational and environmental factors (noise, altitude, hypoxia, acceleration forces, tobacco smoking, oral health and stress) regarding their importance for CVDs risk in military professionals and military aviators.

The title describes the core message of the paper.

The structure of the paper is accurate.

Conclusions are consistent with the evidence and arguments presented.

However, I have same important suggestion regarding this paper.

1. The abstract include only information about introduction and aims. Please add to this section informations regarding results and conclusions.

2. Some figures, diagrams or tables summarising cited literature, which was found during literature searching process will be very helpful for readers and add an additional value for this manuscript.

3. It's seems that there is lack information about the number of included papers to this review.

Author Response

Reviewer 2

Dear Editor and Authors,

I read the review paper entitled "Selected exogenous (occupational and environmental) risk factors of cardiovascular diseases in military and aviation". The topic is original and manuscript discuss the influence of selected occupational and environmental factors (noise, altitude, hypoxia, acceleration forces, tobacco smoking, oral health and stress) regarding their importance for CVDs risk in military professionals and military aviators.

The title describes the core message of the paper.

The structure of the paper is accurate.

Conclusions are consistent with the evidence and arguments presented.

We would like to express our gratitude to the Reviewer 2 for their time and effort spend for reviewing of our manuscript and for all suggestions. All of them have been addressed below and proper modification have been introduced to manuscript (green font). We really hope The Reviewer 2 will find them suitable and sufficient.

However, I have same important suggestion regarding this paper.

  1. The abstract include only information about introduction and aims. Please add to this section informations regarding results and conclusions.

As suggested by the Reviewer 2, these information has been included.

  1. Some figures, diagrams or tables summarising cited literature, which was found during literature searching process will be very helpful for readers and add an additional value for this manuscript.

We really appreciate this suggestion. As this remark was shared by both of the Reviewers, proper summary figure has been included.

  1. It's seems that there is lack information about the number of included papers to this review.

We would like to explain, that in the present version of our manuscript 83 sources of 280 found has been cited and 82 of them were published after year 2000. This information is given in ‘materials and methods’ section.

Round 2

Reviewer 2 Report

Comments and Suggestions for Authors

Dear Editor and Authors,

The manuscript was substantially improved. 

Eventually, more Figures and Tables that would be helpful to the Readers could be included.

Author Response

Reviewer 2

Dear Editor and Authors,

The manuscript was substantially improved. 

Eventually, more Figures and Tables that would be helpful to the Readers could be included.

Response: We would like to thank Reviewer 2 for acceptance of our improvement as well as for the suggestion of additional tables or figures. As advised, summary table (Table 1) of described exogenous risk factors and their CVDs implications has been included.

Table 1. Summary of exogenous risk factors’ influence on CVDs risk

Risk
factor

Cardiovascular implications

References

Noise

increased susceptibility to cardiovascular events, cardiac disease, hypertension, higher diastolic blood pressure, hypotension, oxidative stress, vascular dysfunction, impaired vasodilation, peripheral vasoconstriction, increased peripheral vascular resistance, arterial wall thickening and stiffness, accelerated progression of atherosclerosis, elevated blood glucose levels, increased heart rate and blood pressure, high resting pulse rate,

20; 30; 32; 33; 35; 39

Altitude and hypoxia

lower level of blood or tissue oxygenation, pulmonary barotrauma and cerebral arterial gas embolism, alterations in heart rate and mean blood pressure, enhanced sympathetic tone and pulmonary artery vasoconstriction, reductions in peripheral arterial oxygen saturation and cerebral oxygenation, increased blood pressure and heart rate, acute reduction in cerebral regional oxygen saturation, elevated HDL levels, increased risk of hypertension, changes in the oxygen partial pressure in the arterial blood perfusing the brain, modulation of the sympathetic and parasympathetic cardiac activity, decrease in the partial pressure of oxygen of arterial blood, stimulation of peripheral chemoreceptors in the carotid and aortic arteries, regulation of vascular resistance, arterial pressure, heart rate, cardiac output, and myocardial contractility, increased heart rate and systolic blood pressure, tachycardia, decreases or increases in diastolic blood pressure, significant increase in mean R-R interval values and a slight indirect downward trend in sympathetic cardiac activity and diastolic pressure values

42; 49; 50; 51

Acceleration forces

intravascular contracture, decreased oxygen-carrying capacity, decrease in heart rate, abnormal sympathetic autonomic responses, modulation of membrane fluidity, increased levels of blood serum TAG, increase in HDL levels

53;54; 55

Stress

effects on blood pressure, effect on cardiac function, influence on endothelial functioning, platelets, and metabolic function, regulation of heart rate variability, low heart rate, non-fatal myocardial infarction, CVDs death, increased risk for all types of stroke including fatal stroke, greater general risk of CVD, increased risk for coronary artery disease and related mortality, higher incidence of arterial hypertension, cardiac arrhythmias and myocardial infarction, hypertension, elevated total cholesterol levels, elevated LDL, decreased HDL, elevated TAG levels, elevated homocysteine levels, changes in receptors that regulate blood pressure, allostatic overload, platelet activation, inflammation, arterial hypertension, central obesity, insulin resistance, increased release of inflammatory cytokines involved in platelet activation and endothelial dysfunction, elevated blood pressure

58; 59; 63; 64; 65; 66

Tobacco smoking

increased oxidative stress through generation of ROS, endothelial dysfunction, oxidation of both HDL and LDL, reverse cholesterol transport, elevated plasma fibrinogen levels and inflammatory markers levels (C-reactive protein (CRP) and interleukin-6 (Il-6), increase in the inflammatory process, changes in the thrombosis/fibrinolysis system, hypercholesterolemia, decreased HDL levels, increased blood pressure, increased blood tendency to clotting, increased heart rate, enhanced excretion of thromboxane, which results in increased vasoconstrictive properties of smoking, increased sympathetic activities and plasma catecholamine levels, elevated plasma noradrenaline levels, increased resting heart rate and decreased heart rate variability, higher total cholesterol levels,

9; 70; 72; 74; 80;

Oral health

production of inflammatory mediators such as: tumor necrosis factor α (TNF-α), interleukins (IL-1, IL-6, IL-8) and CRP, presence of Porphyromonas gingivalis, Aggregatibacter actinomycetemcomitans and Streptococcus sanguis in patch of atheroma

82;